# Glucocorticoids and Their Receptor Isoforms: Roles in Female Reproduction, Pregnancy, and Foetal Development

**DOI:** 10.3390/biology12081104

**Published:** 2023-08-09

**Authors:** Sreeparna Bhaumik, Jack Lockett, James Cuffe, Vicki L. Clifton

**Affiliations:** 1Mater Research Institute, Faculty of Medicine, The University of Queensland, Brisbane 4067, Australia; s.bhaumik@uqconnect.edu.au (S.B.); jack.lockett@health.qld.gov.au (J.L.); 2Department of Diabetes and Endocrinology, Princess Alexandra Hospital, Metro South Health, Brisbane 4102, Australia; 3School of Biomedical Sciences, The University of Queensland, Brisbane 4067, Australia; j.cuffe1@uq.edu.au

**Keywords:** glucocorticoid receptor, reproduction, pregnancy, stress

## Abstract

**Simple Summary:**

This review explores how glucocorticoids influence female fertility, reproduction, and foetal development. Glucocorticoids interact with specific receptors located in various parts of the reproductive system, potentially altering their function and impacting fertility. When a pregnant woman is exposed to stress or excess glucocorticoids, these effects can be long-lasting and alter the growth and development of the baby. Interestingly, there are different versions of the glucocorticoid receptor found in different parts of the reproductive system and the placenta. These various receptor types could explain why there is a range of responses to stress, different pregnancy outcomes, and even why males and females respond differently to changes in glucocorticoid concentrations. This review sums up our current knowledge about these different receptor types and their roles in hormone signalling, reproduction, and foetal growth.

**Abstract:**

Alterations in the hypothalamic–pituitary–adrenal (HPA) axis and associated changes in circulating levels of glucocorticoids are integral to an organism’s response to stressful stimuli. Glucocorticoids acting via glucocorticoid receptors (GRs) play a role in fertility, reproduction, placental function, and foetal development. GRs are ubiquitously expressed throughout the female reproductive system and regulate normal reproductive function. Stress-induced glucocorticoids have been shown to inhibit reproduction and affect female gonadal function by suppressing the hypothalamic–pituitary–gonadal (HPG) axis at each level. Furthermore, during pregnancy, a mother’s exposure to prenatal stress or external glucocorticoids can result in long-lasting alterations to the foetal HPA and neuroendocrine function. Several GR isoforms generated via alternative splicing or translation initiation from the GR gene have been identified in the mammalian ovary and uterus. The GR isoforms identified include the splice variants, GRα and GRβ, and GRγ and GR-P. Glucocorticoids can exert both stimulatory and inhibitory effects and both pro- and anti-inflammatory functions in the ovary, in vitro. In the placenta, thirteen GR isoforms have been identified in humans, guinea pigs, sheep, rats, and mice, indicating they are conserved across species and may be important in mediating a differential response to stress. Distinctive responses to glucocorticoids, differential birth outcomes in pregnancy complications, and sex-based variations in the response to stress could all potentially be dependent on a particular GR expression pattern. This comprehensive review provides an overview of the structure and function of the GR in relation to female fertility and reproduction and discusses the changes in the GR and glucocorticoid signalling during pregnancy. To generate this overview, an extensive non-systematic literature search was conducted across multiple databases, including PubMed, Web of Science, and Google Scholar, with a focus on original research articles, meta-analyses, and previous review papers addressing the subject. This review integrates the current understanding of GR variants and their roles in glucocorticoid signalling, reproduction, placental function, and foetal growth.

## 1. Glucocorticoids and the Hypothalamic–Pituitary–Adrenal Axis

Glucocorticoids, named for their effects on glucose metabolism, play a crucial role in regulating metabolic homeostasis, the cardiovascular system, cell proliferation and survival, growth, cognition and behaviour, immune function, and reproduction [1]. Glucocorticoids are steroid hormones that are synthesised and released by the adrenal cortex under the regulation of the hypothalamic–pituitary–adrenal (HPA) axis (Figure 1) [2]. In the presence of psychological or physiological stress, hypothalamic neurons in the paraventricular nucleus (PVN) synthesise and release corticotropin-releasing hormone (CRH) into the hypophysial portal vessels, which stimulates the release of adrenocorticotropic hormone (ACTH) from the anterior pituitary into the bloodstream [3]. ACTH primarily acts on the adrenal cortex, stimulating the synthesis and secretion of glucocorticoids from the zona fasciculata [2]. Glucocorticoids can inhibit CRH expression and secretion and ACTH output via negative feedback mechanisms [4,5]. Under normal, unstressed conditions, glucocorticoids are released in a circadian and ultradian rhythm. Exposure to acute stressors activates the HPA axis and leads to a temporary increase in glucocorticoids, which returns to baseline following resolution of the stressor; conversely, chronic stress can lead to disruption of the circadian rhythm and derangement of the ultradian rhythm of glucocorticoid secretion [2,6]. While glucocorticoids are typically adaptive to stress, excessive or inadequate activation of the HPA axis may contribute to the development of pathological conditions [6].

## 2. Functions of Glucocorticoids

Glucocorticoids play vital roles in a wide range of physiological processes crucial for maintaining homeostasis in the body. They upregulate enzymes involved in gluconeogenesis and glycogen synthesis in the liver and inhibit glucose uptake and utilisation in the muscle and adipose tissues [7,8,9]. This not only helps maintain glucose homeostasis during normal times but also, during periods of stress such as starvation or physical exertion, ensures that glucose remains the primary source of energy for vital organs such as the brain and skeletal muscles. Cortisol is also involved in sodium and water homeostasis to ensure sufficient blood volume and the maintenance of appropriate blood pressure [10,11]. It also plays an important role in the nervous system, with increased levels of glucocorticoids associated with various mental health disorders, including schizophrenia, post-traumatic stress disorder (PTSD), anxiety, depression, and addiction to drugs [12,13,14]. Prior to birth, glucocorticoids are recognised for their role in promoting the maturation of foetal lungs [15,16], but they can also lead to the development and differentiation of other essential organs [17].

The immunomodulatory actions of glucocorticoids are highly complex and include both immune stimulatory and immunosuppressive actions. Glucocorticoids are widely prescribed for their anti-inflammatory and immunosuppressive properties to treat inflammatory conditions, autoimmune disorders, haematological malignancies, and many other diverse uses [18]. Physiologically, glucocorticoids enhance innate immunity while limiting inflammation, supporting pathogen clearance, and preventing excessive tissue damage, to restore homeostasis following an inflammatory response [19]. These dichotomous impacts on inflammation are dictated by the cellular lineage and activation state, the stage of the inflammatory response, and glucocorticoid concentration [20]. In the innate immune response, glucocorticoids upregulate pattern-recognition receptors such as Toll-like receptor 2 (TLR2) and NOD-like receptors (NLRP1, NLRP3, and NLRC4) [21] to improve detection of pathogen-associated or damage-associated molecular patterns (PAMP/DAMPs). Downstream of these receptors, pro-inflammatory cytokines (such as IL-6, IL-1β, and TNF-α) are produced under the control of transcription factors nuclear factor kappa-light-chain-enhancer of activated B cells (NF-κB) and activator protein 1 (AP-1) [19,22]. These pro-inflammatory cytokines initiate an immune and acute-phase response that promotes glucocorticoid secretion, creating a feedforward mechanism that limits inflammation [22]. Glucocorticoids directly inhibit AP-1 and NF-κB, and upregulate their inhibitors, effectively dampening the inflammatory response and contributing to the resolution of inflammation [19]. Glucocorticoids further influence the resolution phase by inhibiting neutrophil recruitment, promoting apoptosis in neutrophils and lymphocytes, and activating macrophages at the injury site for tissue repair [19]. However, when administered in therapeutic doses, glucocorticoids primarily exhibit immunosuppressive effects. These effects include inhibiting leukocyte migration into tissues, inducing eosinophil apoptosis and preventing neutrophil emergence from the marrow, reducing circulating lymphocyte counts (more so T cells than B cells) through apoptosis and sequestration, and subsequently reducing immunoglobulin levels [23,24,25,26].

## 3. Glucocorticoid Bioavailability

A number of factors can impact glucocorticoid bioavailability. Roughly 95% of circulating glucocorticoids are bound to cortisol-binding globulin (CBG) or albumin following secretion, and the expression of these binding proteins impacts the level of freely available, biologically active glucocorticoids [27]. Endogenous glucocorticoids in circulation have an inconsistent half-life (66–120 min); however, those bound to CBG tend to have a longer half-life compared to the unbound hormone [27].

Given the lipophilic nature of glucocorticoids, free hormones can traverse the plasma membrane through simple diffusion. In cells that express multi-drug resistance p-glycoprotein, intracellular glucocorticoid concentration can be limited through their active expulsion [2]. Once inside the cell, glucocorticoids’ conversion to active forms (in humans, cortisone to cortisol) by the enzyme 11β-HSD 1 or to inactive forms (cortisol to cortisone) by 11β-HSD 2 regulates the availability of ligands to the glucocorticoid receptor (GR) [2]. Through these mechanisms, glucocorticoid availability to activate GR and hence downstream signalling can be finely tuned [2].

## 4. Glucocorticoid Receptors

The cellular action of glucocorticoids is mediated by an intracellular protein, the glucocorticoid receptor (GR) [28]. A number of GR isoforms, originating from different splice variants and transcription initiation sites, have been discovered. These isoforms diversify the range of glucocorticoid actions and play a role in determining cellular responsiveness to glucocorticoids [29]. While not the only determinant of glucocorticoid action, the composition of different glucocorticoid receptor isoforms differs between cell types and likely mediates differences in cellular activation state. Furthermore, differences in cellular responses to high or low glucocorticoid doses, natural or synthetic glucocorticoids, as well as changes that occur over time, are likely mediated by glucocorticoid receptor subtypes.

The GR is a member of the highly conserved nuclear receptor subfamily 3 [30]. All members of this superfamily share a similar domain structure, consisting of an N-terminal ligand-independent transactivation (AF1) domain, a central DNA-binding domain (DBD), and a C-terminal ligand-binding domain (LBD) [31]. Apart from containing the ligand-binding pocket, the LBD also contains crucial sequences for receptor dimerisation and nuclear localisation, as well as a second transactivation domain (AF2) that facilitates interactions with coregulators in a ligand-dependent manner [31]. A small flexible hinge region separates the DBD and LBD and is involved in conformational changes induced after ligand-binding [31].

The human GR, encoded by the NR3C1 gene, is composed of nine exons [31]. Exon 1 forms the 5′-untranslated region, while exon 2 encodes the N-terminal domain (NTD), which is poorly conserved across nuclear receptors [31]. Exons 3 and 4 constitute the DBD, the most conserved domain in the nuclear receptor family, while exons 5–9 encode the hinge region and ligand-binding domain (LBD) [31]. The NR3C1 gene undergoes alternative splicing, leading to the production of human GRα, GRβ, GRγ, GR-A, and GR-P transcriptional isoforms (Figure 2) [31,32,33]. Most research has focused on glucocorticoid signalling through GRα and GRβ, proteins identical up to amino acid 727 but differing in their C-terminal exon 9 [34]. The GRβ variant lacks sequences encoding the 11th and 12th helices of the LBD, making it incapable of binding glucocorticoids [34]. Though initially thought to serve primarily as a dominant negative regulator of GRα, transcriptional analysis shows that GRβ has inherent transcriptional activity when overexpressed [35,36,37].

The sequence of GR-γ differs due to an arginine residue included between exons 3 and 4 of the DBD, resulting from the use of an alternative intronic splice donor site [38]. The insertion of a single amino acid significantly reduces the transcriptional activity of GR-γ, demonstrating the functional significance of this region [39,40]. Moreover, changes in the DBD of GR-γ alter its DNA binding sequence specificity, leading to a unique transcriptome [41,42]. The in vivo functions of the GR-A and GR-P isoforms, which contain incomplete LBDs, are not well understood, but both have been demonstrated in glucocorticoid-resistant malignant cell lines [33,43].

All GR splice variants could possibly generate additional isoforms through alternative translation initiation mechanisms. However, these translational isoforms have only been described for GRα to date. In exon 2, eight highly conserved AUG start codons generate various GR isoforms through ribosomal leaky scanning and shunting, including GRα-A, -B, -C1, -C2, -C3, -D1, -D2, and -D3 (Figure 2). The relative expression of both splice and translation GR isoforms can vary among tissues and within individual cells, influencing their transcriptional potential [44]. Each GR isoform is also subject to several post-translational modifications, such as phosphorylation, acetylation, ubiquitination, and sumoylation, further influencing receptor activity [1].

**Figure 2 biology-12-01104-f002:**
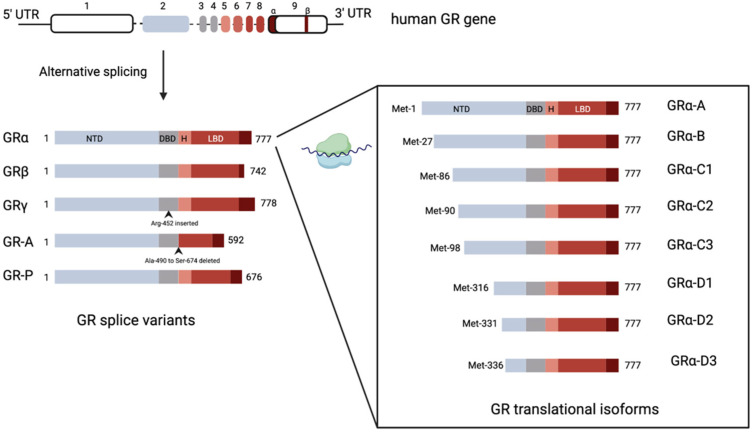
Splice and translation initiation isoforms of the glucocorticoid receptor. Exons of the glucocorticoid receptor gene with the subsequent splice (left) and translation initiation (right) isoforms. AF-1—transcriptional activation function 1; DBD—DNA binding domain; GR—glucocorticoid receptor; LBD—ligand-binding domain; NTD—N-terminal domain. Created on bio render. Adapted from Ramamoorthy and Cidlowski [45].

The traditional explanation for GR signalling focused on glucocorticoids passing through the plasma membrane and binding to GR anchored within the cytoplasm, which induced translocation to the nucleus to regulate gene transcription. More recent discoveries have expanded our understanding of GR function to include a multitude of mechanisms by which GRs impact gene regulation, the varying impact of GR isoforms on regulating cellular response, and non-genomic effects of GR. There are methodological limitations in identifying and understanding the effects of individual GR isoforms. These limitations include a lack of specific antibodies for each GR isoform, leading to the use of the semi-quantitative technique of Western blot for the characterisation of the GR isoform profile in different cell types and tissues, distinguishing different isoforms by molecular weight. Furthermore, identifying the role of each specific GR isoform has been limited to overexpression experiments in GR null cell lines [44], which does not account for the interaction of multiple GR isoforms that are expressed endogenously.

## 5. Glucocorticoid Receptor Signalling

More recent understanding of glucocorticoid function has demonstrated that they act through both genomic (predominant) and non-genomic pathways. The genomic pathway is mediated by GRs (Figure 3). In the absence of glucocorticoids, GRs exist principally anchored in the cytoplasm as part of a large complex of proteins, including chaperones (hsp90, hsp70, and p23) and immunophilins (FKBP51 and FKBP52), which help maintain GRs in a conformation that allows them to bind to ligands with high affinity [46]. Once glucocorticoids bind to GRs, receptor conformational changes lead to the dissociation of the protein complex and the exposure of 2 nuclear localisation signals [45]. The ligand-bound GR is then quickly transported into the nucleus through nuclear pores. Inside the nucleus, the GR binds directly to glucocorticoid response elements (GREs) as a dimer and activates the expression of target genes [45]. This binding leads to further conformational changes in the GR, allowing it to recruit coregulators and chromatin-remodelling complexes that influence the activity of RNA polymerase II and activate gene transcription and repression [45]. Binding to negative glucocorticoid-responsive elements (nGREs) can recruit corepressors (NCoR1 and SMRT) and histone deacetylases (HDACs) to repress the activity of target genes [47]. In addition to its direct interaction with GREs, the GR can also interact with members of the signal transducer and activator of transcription (STAT) family to enhance transcription of specific target genes [47]. The anti-inflammatory effects of glucocorticoids are mainly mediated by a negative regulatory mechanism called trans-repression. In this, the ligand-bound GR is recruited to genes by directly interacting with DNA-bound transcription factors, particularly NF-κB and activator protein-1 (AP-1) [48]. There are a number of proteins within the cell that can interact with the GR and potentiate trans-repression. For instance, O-linked-N-acetylglucosamine transferase (OGT), a placental stress biomarker, can interact with GRs and impact NF-kB-mediated transcription [49]. The GR directly binds to the p65 subunit of NF-kB and the Jun subunit of AP-1, inhibiting their transcriptional activation [48]. For some genes, GRs function in a composite manner, binding directly to a GRE and physically associating with AP1 or NF-kB bound to a neighbouring site on the DNA, while for others, GRs bind to transcription factors without interacting directly with DNA [45].

The non-genomic effects of glucocorticoids, which occur without requiring gene transcription, are rapid and mediated through physiochemical interactions of cytosolic or membrane-bound GRs [50]. These effects happen within seconds to minutes of activation and involve the activity of kinases such as phosphoinositide 3-kinase, AKT, Src kinase, mitogen-activated protein kinases (MAPKs), or intracellular Ca^2+^ signalling [51,52,53,54]. Glucocorticoids can non-genomically interact with membrane lipids and proteins, cytoplasmic proteins, including various kinases, transcription factors, and glucocorticoid transporters, to influence cell function [53]. High doses of glucocorticoids can directly interact with membrane lipids to alter membrane fluidity and affect ATP use and ion cyclicity [55]. Glucocorticoids can also enhance Ca^2+^ signalling by binding to voltage-dependent calcium channels in brain synapses. For example, in cultured rat hippocampal cells, stress-induced elevated levels of corticosterone rapidly prolonged NMDA receptor-mediated Ca^2+^ signalling via the activation of a protein kinase C-dependent process by membrane GRs [56]. Similarly, glucocorticoids have been found to non-genomically impact neurotransmission by affecting GABA, glutamate, serotonin, acetylcholine, and vasopressin receptors [57,58,59,60]. Furthermore, glucocorticoids can engage with a variety of cytoplasmic proteins, including members of the MAPK family (extracellular signal-regulated protein kinases (ERK), c-Jun NH2-terminal protein kinase (JNK), and MAPK p38), phospholipases, and protein kinases, both genomically and non-genomically. MAPKs can regulate processes such as cell proliferation, development, differentiation, and apoptosis [61,62]. Some examples of these interactions in relation to reproductive organs will be discussed in the forthcoming sections. Glucocorticoid administration to mouse thymocytes induced apoptosis via activation of phospholipase C, non-genomically, in vitro [63]. Its administration to A549 cells in vitro prevented cell growth due to inhibition of prostaglandin E2 via a reduction in the activity of phospholipase A [64]. Glucocorticoids can also secondarily activate MAPKs [65], Ca^2+^ [66], and nitric oxide (NO) [67] signalling and impact ion transport [68] by activating protein kinases A, B, and C. Glucocorticoids can bind the extracellular site of membrane-bound GRs, which can then interact with and activate the stimulatory G protein, G_αs_, to rapidly increase the levels of 3′,5′-cyclic adenosine monophosphate (cAMP), a common secondary messenger involved in the regulation of various biological processes and the actions of non-steroidal hormones [69,70]. Lastly, when glucocorticoids bind to GRs, components of the multi-protein anchoring complex are released, which participate in secondary signalling pathways. The accessory proteins that have been reported to mediate these non-genomic effects of glucocorticoids include HSP-70, HSP-90, and MAPKs such as Src [71,72]. Moreover, the ligand-bound GR can also interact with other proteins, including ZAP-90, leading to further downstream signalling [73].

This non-genomic signalling of the glucocorticoid receptor increases the complexity and diversity of biological actions that are dependent on glucocorticoids [45].

## 6. Glucocorticoids on Hypothalamic–Pituitary–Gonadal Axis

The hypothalamus and pituitary are also central to reproductive physiology, serving as hubs for both receiving and delivering endocrine signals [74]. The hypothalamus generates and releases gonadotropin-releasing hormone (GnRH), which travels through the hypophyseal portal system to reach the anterior pituitary [75]. There, GnRH stimulates the synthesis and release of follicle-stimulating hormone (FSH) and luteinising hormone (LH) from the gonadotrophs, leading to the secretion of estradiol and progesterone from the ovaries [76]. A regulatory negative feedback mechanism exists in which estradiol from the ovaries inhibits the secretion of GnRH and thus the production of FSH and LH [76]. Increased glucocorticoid exposure, whether resulting from stress or exogenous administration, can lead to significant reproductive disturbances through their notable impacts on the hypothalamus and pituitary [77,78,79]. Glucocorticoids influence the HPG axis by inhibiting the release of GnRH from the hypothalamus and the synthesis and release of gonadotropins from the pituitary [77,79].

The presence of GR in the hypothalamus implies that glucocorticoids could have direct effects [80]. Indeed, dexamethasone-induced GR signalling in hypothalamic cell lines that secrete GnRH suppresses GnRH promoter activity and subsequent GnRH mRNA levels in vitro by interacting with a multiprotein complex at negative GRE sites [81]. A study analysing the reproductive function in adult male rats treated with corticosterone over an extended period reported decreased hypothalamic GnRH mRNA and decreased serum LH but not FSH secretion [82]. The same study examining the acute effects of dexamethasone on reproductive function in adult female rats reported a significant decrease in pituitary FSHβ but not LHβ mRNA levels [81]. The findings from this study suggest that chronic increases in levels of glucocorticoid/stress suppress gonadotropin levels by reducing the expression of GnRH levels, while acute glucocorticoid administration prevents gonadotropin secretion via other processes, including the reduction of FSHβ mRNA levels. However, this effect of acute stress and short-term increases in glucocorticoid levels on gonadotrophin secretion can vary, either inhibiting or stimulating it. GnRH stimulates the production and release of LH and FSH by binding to the GnRHR in pituitary gonadotropes [83,84]. Evidence from the mouse pituitary gonadotrope cell line LβT2 treated with dexamethasone suggests that glucocorticoids could directly induce the expression of the GnRHR gene, leading to increases in LH and FSH secretion [85]. In addition, in primary cultured cells from the anterior pituitary, glucocorticoids have been found to increase FSHβ mRNA levels [82,86]. These are inconsistent with the findings from Gore et al. [82], which may suggest that the way glucocorticoids control the expression of gonadotropin synthesis is dependent on the context. Our understanding of the regulation of gonadotrophins by glucocorticoids is presently based on findings from cell line studies, which may not accurately depict the regulation by endogenous glucocorticoids in vivo [79].

Besides directly controlling GnRH, GnRHR, and pituitary gonadotropes, glucocorticoids can also regulate their inhibitory effects on HPG function via novel intermediaries. Glucocorticoids can inhibit GnRH expression by regulating the expression of two neuropeptides, kisspeptin (KISS1) and gonadotropin-inhibitory hormone (GnIH) [79]. These neuropeptides exert contrasting impacts on GnRH secretion from the hypothalamus and react to elevated glucocorticoid levels. KISS1 promotes GnRH secretion via its receptor, KISS-1R, which is co-expressed in GnRH neurons. KISS1 neurons located in the anteroventral periventricular nucleus and the periventricular nucleus continuum of the preoptic area in the hypothalamus express GRs, indicating that glucocorticoids can directly act on these neurons [87]. Corticosterone administration in mice inhibited ovarian cyclicity by disrupting the preovulatory hormonal cascade required for the LH surge [88]. The study also reported a significant decrease in the activation and percentage of KISS1 neurons as well as the expression of KISS1 mRNA in the hypothalamus during an estradiol-induced LH surge [88]. This suggests that glucocorticoids can suppress the HPG axis by inhibiting the expression of KISS1 neurons. GnIH neurons suppress the functions of GnRH and KISS1 neurons and gonadotrophs [89]. Both acute and chronic stress in sheep led to an increase in the expression and activity of GnIH neurons and increased the interaction between GnIH fibres and GnRH cells [90]. However, stress had no effect on GnIH levels in the hypophysial portal blood [90]. Moreover, this effect on GnIH gene expression disappeared after adrenalectomy, implying that adrenal-derived factors, most likely corticosterone, mediated the stress-induced inhibition of reproductive functions [90]. GnIH neurons co-express GR, and the exposure of quail and rat hypothalamic neuronal cells to corticosterone resulted in increased GnIH expression [90]. This increase was facilitated through glucocorticoid response elements (GREs) in the promoter region of GnIH [90]. In short, glucocorticoids can indirectly influence the reproductive functions of the HPG axis by regulating GnRH expression and secretion by KISS1 and GnIH.

## 7. Effect of Glucocorticoids on Reproductive Organs

As glucocorticoids are known to be a key regulator of basal and stress-related homeostasis, it is highly likely that stress-related increases in glucocorticoid levels could contribute to decreased fertility [77]. Stress-related compromises of reproductive function can be reported across various species. Mammals such as rats [91,92,93], along with birds and reptiles [94,95,96,97], all exhibit a reduction in both male and female reproductive abilities in response to physical stress [3]. The response to stress in humans is also detrimental to reproductive function throughout the life cycle. For example, high perceived stress during pregnancy is a risk factor for preterm labour and poor outcomes in the offspring [98,99]. For female athletes, there is a high prevalence of delayed puberty, menstrual disorders, and long-term secondary amenorrhoea, especially among those participating in aesthetic sports and endurance sports [100,101]. This reproductive dysfunction could be caused by the alteration of HPG axis activity due to excess cortisol associated with physiological and psychological stress on the body. Glucocorticoids are also known to interfere with sexual maturation, as patients with hypocortisolism (Addison’s disease) and hypercortisolism (Cushing’s syndrome) have altered onsets of puberty [102,103]. A study revealed a positive correlation between circulating glucocorticoid levels and the age of puberty onset in prepubertal girls [103]. This is consistent with findings from animal studies [104,105,106]. Studies in mice and rats report delayed onset of puberty in offspring of mothers subjected to stress or with synthetic glucocorticoid administration [106,107,108]. Prepubertal exposure to synthetic glucocorticoids delayed the onset of vaginal opening, an indicator of puberty in female rats [109].

### 7.1. Ovary

Apart from influencing the ovarian cycle via the hypothalamus and pituitary, glucocorticoids can also directly affect ovarian functions by controlling the activities of granulosa cells, oocytes, cumulus cells, and luteal cells through GRs [88]. It was previously thought that the ovary did not produce glucocorticoids locally. Hence, it was presumed that glucocorticoids in the ovary were transported from the adrenal glands via the bloodstream [110]. These glucocorticoids, probably inactive (cortisone), were then proposed to diffuse into the granulosa cells of the periovulatory follicle, where 11β-HSD 1 converted them to cortisol, which could then exert its effects via co-localised GRs [110]. However, a recent study by Jeon et al. [111] demonstrated that human primary luteinising granulosa cells produce cortisol in vitro. The mRNA levels of CYP11B1 and CYP21A2, two enzymes required for the biosynthesis of cortisol, were detected in primary cultured human granulosa/lutein cells (hGLC), and their levels were found to increase following ovulatory administration of human chorionic gonadotropin (hCG) as a substitute for LH [111]. CYP21A2, an enzyme necessary for the metabolism of 17α-hydroxyprogesterone into 11-deoxycortisol, and CYP11B1, an enzyme responsible for the conversion of 11-deoxycortisol into cortisol, are both required for the synthesis of cortisol from progesterone (Figure 4) [111]. This is consistent with findings from Amin et al. [112], who reported the levels of two cortisol precursors, 11-deoxycorticosterone and 11-deoxycortisol, in the follicular fluid to be positively correlated with the lipid content in human luteinised granulosa cells (LGCs) and demonstrated the expression of CYP21A2 mRNA in LGCs in vitro. Given that during ovulation, human granulosa cells generate a substantial amount of progesterone, the presence of enzymes that convert progesterone to cortisol suggests that these cells could produce cortisol [111]. However, it is still likely that the majority of the cortisol acting on the ovaries is derived from the adrenal glands instead of being locally produced, as this evidence is based on in vitro studies and might not accurately represent in vivo physiology.

Glucocorticoid concentrations are intracellularly regulated in the ovary by 11β-HSD 1 and 2. 11β-HSD has been shown to be present in various ovarian cell types such as oocytes, cumulus cells, granulosa cells, theca cells, granulosa-lutein cells, corpus luteum, and ovarian surface epithelium [113,114,115]. During the LH surge, there is a rapid increase in the production of progesterone and cortisol. Before the LH surge, follicular levels of inactive glucocorticoids (cortisone) were higher than those of cortisol [116]. However, post-surge, this reverses, with the concentration of cortisol being 4.5-fold higher than that of cortisone [116]. This switch is proposed to be caused by a change in the predominance of the 11β-HSD isotype from 2 to 1 [116,117]. In non-luteinised granulosa cells from IVF patients, 11β-HSD 2 was readily detected, but it was not detectable in the luteinising granulosa cells [118]. More recent studies have also demonstrated a swift decrease in 11β-HSD 2 (mRNA) and increased 11β-HSD 1 (mRNA and protein) levels in human granulosa cells collected after hCG stimulation in comparison to cells obtained prior to the in vivo LH surge [119], as well as in vitro [111]. The increased expression of 11β-HSD 1 persisted in post-ovulatory follicles, suggesting it has a role during ovulation and the initial phase of luteal formation by modulating intracellular glucocorticoid concentrations. Furthermore, 11β-HSD 1 expression was mainly observed in granulosa cells from late ovulatory follicles and granulosa-lutein cells of post-ovulatory follicles, indicating the location of cortisol synthesis [111]. These findings suggest that active glucocorticoid concentrations are reduced during follicular growth and maturation, but they rise during the ovulation process initiated by the LH surge.

Preovulatory follicles undergo an acute inflammatory event during ovulation that is associated with rupture, the subsequent healing process, and metabolic alterations linked with luteinisation. Thus, the coinciding increase in expression of anti-inflammatory glucocorticoids [120] by 11β-HSD 1 activation of cortisone to cortisol may be a physiological compensatory mechanism to reduce the ovarian inflammatory process [111]. To determine the effect of cortisol on periovulatory follicles, Jeon et al. [111] looked at the effect of ovulatory hCG stimulation on the expression of genes involved in cortisol synthesis. The expression of 11β-HSD 1, NR3C1, and FKBP5 was increased after hCG administration in periovulatory follicles, both in vitro and in vivo [111]. Moreover, it was reported through immunohistochemistry that GR was localised in endothelial cells and leukocytes within and near the late and post-ovulatory follicles [111]. These findings suggest that glucocorticoids probably operate during ovulation and the initial stages of corpus luteum formation by exerting their effect directly on follicular cells, endothelial cells, and leukocytes, all of which express GR. Since cortisol plays a role in ovulation and corpus luteum (CL) formation, this could contribute to the fact that women with dysregulated levels of cortisol frequently experience menstrual irregularities and anovulatory infertility.

Furthermore, stress-induced glucocorticoid levels have been linked to a decrease in oocyte competence. Chronic heat stress in pigs has been shown to modify ovarian expression of genes involved in steroidogenesis and increase signalling via the insulin-mediated phosphoinositide 3-Kinase (PI3K) pathway [121]. Heat stress can also alter the composition of follicular fluid, thereby changing the oocyte’s microenvironment [122,123]. In mice, the damage to oocyte competence was found to be dependent on the severity of the restraint stress, with more significant effects observed after prolonged stress [124]. Administration of exogenous glucocorticoids in female rats led to a decline in growth factor levels and alterations in the estrogen-to-progesterone ratio [125].

#### 7.1.1. Dual Role of Glucocorticoids in the Ovary

Despite the studies aimed at determining the role of glucocorticoids in the ovary, their function is not completely understood, with contradictory findings from multiple studies. Several studies have looked at the effect of glucocorticoid administration on mammalian oocytes and its influence on oocyte maturation, with inconsistent results. In some in vitro mouse studies, no effect was found on oocyte maturation when they were subjected to cortisol (up to 28 μM) and dexamethasone (up to 100 μM) [126,127,128,129]. However, in other mouse studies, slightly higher concentrations of cortisol (138 μM) and dexamethasone (204 μM) resulted in inhibition of oocyte maturation in vitro [126,128,129]. Similar to studies in mice, cortisol and dexamethasone administration (between 0.26 and 138 μM) inhibited oocyte maturation in porcine oocytes in vitro [130]. Conversely, a study on horses reported similar doses of cortisol (0.28–2.8 μM) to have no effect on oocyte maturation in vitro [131]. Glucocorticoids can either selectively induce or inhibit ovarian steroidogenesis. Dexamethasone administration in rats is associated with altered production of progesterone from granulosa cells by either increasing or decreasing the expression of steroidogenic proteins like steroidogenic acute regulatory protein (StAR) [132,133,134]. This indicates that in vitro glucocorticoids exhibit both stimulatory and inhibitory effects in the ovary. Such discrepancies could be attributed to variations in the administered dosage, the phase of follicular development, or the differential expression of GR isoforms.

Similarly, glucocorticoids can have both pro- and anti-inflammatory roles in the ovary [135]. Cortisol treatment has been shown to decrease germ cell density by increasing the apoptosis of oogonia in the developing human ovary, impair oocyte development potential, and increase the apoptosis of granulosa cells in animal models [136]. However, glucocorticoids have also been reported to inhibit apoptosis and promote survival in ovarian granulosa cells by non-genomically phosphorylating ERK 1 and 2, and Akt/protein kinase B [137,138]. In primary bovine luteal cells, cortisol has been reported to prevent apoptosis induced by TNF and interferon γ (IFN-γ) by inhibiting the expression of caspase 8 and 3, thus preserving the function of the corpus luteum [139]. The anti-apoptotic effects of glucocorticoids could be a mechanism through which glucocorticoids exert their anti-inflammatory effects in the ovary, as they induce the death of pro-inflammatory cells and promote the survival of cells undergoing the inflammatory response. These anti-apoptotic mechanisms could also contribute to reducing the damage associated with follicle rupture. Glucocorticoids, via GR signalling, reduce the expression and function of extracellular matrix metalloproteinase 9 in human ovarian surface epithelial cells, which helps inhibit proteolytic injury to the ovarian surface during ovulation with the associated inflammation [140]. These opposing effects of glucocorticoids on the ovary cannot be explained by the expression of a single isoform of GR and could potentially be a result of differential expression of the GR isoforms.

#### 7.1.2. Glucocorticoid Receptor Isoforms in the Ovary

GRs are found in the follicles, corpus luteum, and cells of the ovarian surface epithelium in both rats and humans [141]. In human studies, GR protein has been identified in ovarian surface epithelium cells, in vitro [142]. As mentioned above, in vivo studies have also found GR protein to be localised in granulosa cells, theca cells, endothelial cells, and leukocytes in blood vessels in the vicinity of late ovulatory and post-ovulatory follicles [111]. The expression levels of GR in the follicle remain unchanged during the stages of follicular maturation and ovulation in rats [143], implying that the main regulatory processes in these cell types could be determined by the concentration of active glucocorticoids. Current evidence proposes that glucocorticoids have a direct influence on the preliminary phases of reproduction, such as the maturation of oocytes, fertilisation, and preimplantation embryo growth [144,145]. However, the observed effects can vary based on factors like the type of glucocorticoid molecule, dosage, stage of development, and species under study [145]. As variation in the expression of GR isoforms may lead to distinct cellular responses to glucocorticoids, investigating the expression of various splice and translational isoforms of the GR in oocytes carries significant importance.

Very few studies have looked at the expression of the different isoforms of the GR in the mammalian ovary. The ovary is a complex organ, with its function highly linked to the active follicles that undergo constant change across the ovarian cycle. This could possibly lead to a constant shift in the expression profiles of the different GR isoforms. A recent study examining mouse oocytes found the two splice variants GRα and GRγ (mRNA) to be highly and consistently expressed, while GRβ and GR-P (mRNA) were undetectable [145]. This study also reported the presence of a doublet band around 94 and 91 kDa in oocytes, and this doublet likely represents mouse isoforms GRα-A and B (GR protein) [145]. Another study [146] examining the presence of GRβ in various tissues reported the absence of GRβ in SK-OV-3 (ovarian adenocarcinoma) cells in vitro, consistent with the results from the study by Cikos et al. [145]. Since GRβ is known to antagonise the effects of GRα, the pro-inflammatory or contradictory effects of cortisol have often been attributed to an increased GRβ:GRα ratio. However, since GRβ is undetectable in the ovary, the varying responses of ovarian cells to high concentrations of glucocorticoids could be attributed to the presence of the other splice or translational isoforms present in the ovary. The detection of various GR isoforms in ovarian cells needs to be further examined to better elucidate their contribution to ovarian glucocorticoid function.

### 7.2. Uterus

The influence of glucocorticoids on uterine biology has been typically depicted as an antagonist of estrogen activity, but it has been recently studied as an independent moderator of uterine functionality. Dexamethasone impedes estrogen-induced uterine growth and proliferation in mice and, when administered before estradiol-facilitated implantation, can lead to a decrease in implantation sites [147,148]. Gene identification and ontology studies looking at the expression of genes regulated by dexamethasone and estradiol in a human uterine epithelial cell line point to glucocorticoids regulating important biological processes, including immune function, cell cycle, and embryonic development [149]. For instance, dexamethasone (but not estradiol) upregulated the expression of nuclear factor of kappa light chain inhibitor alpha (NFκB1A), an inhibitor of the NFκB complex. NFκB signalling in the uterus controls the apoptotic function of caspase 3 during pregnancy [149]. The expression of NFκB1A escalates during the late secretory phase, coinciding with an increased incidence of apoptosis in the uterus [150,151]. Thus, the regulatory role of glucocorticoids on NFκB1A could contribute to the balance between cell death and survival in the uterus [149]. Furthermore, expression of the gene left-right determination factor 1 (LEFTY1) was upregulated with dexamethasone treatment (restored with estradiol administration) [149]. LEFTY1 is a gene that encodes a soluble cytokine of the TGF-β superfamily [152]. When LEFTY1 is introduced into the uterine horn of pregnant mice, it significantly reduces uterine decidualisation and results in a decreased litter size [153]. Thus, glucocorticoid regulation of key fertility genes, like LEFTY1, offers additional evidence that glucocorticoids may function as a bridge between the immune and reproductive systems in the uterus [149].

Dexamethasone administration in mice also singularly regulates the expression of genes associated with cellular development, proliferation, and signalling. In the uterus of neonatal mice, dexamethasone reduces epithelial cell proliferation through mechanisms that are both dependent on and independent of GR [154]. Human endometrial cells that have undergone decidualisation also exhibit a distinct transcriptome dependent on the GR [155]. This GR-dependent transcriptome is enriched with Krüppel-associated box domain containing zinc-finger proteins, which are a family of transcriptional repressors involved in heterochromatin formation [155]. In fact, decidualisation of human endometrial stromal cells was associated with an increase in GR expression [154]. Gene expression analysis of GR knockout HESCs that were undergoing differentiation reported an increase in trimethylated H3K9 levels in decidualising cells [154]. This suggests that GR can directly manage gene expression and globally modify transcription through the regulation of histone alterations in human uterine cells [154].

The semi-allogenic foetus’s immunological tolerance and a successful pregnancy require pre-emptive immune system adjustments that support endometrial receptivity [79]. These changes begin before the implantation of the embryo and escalate throughout the early stages of pregnancy [156,157,158]. During this time, immune cells are mobilised and activated in the uterus, establishing a dynamic immune response that is crucial for implantation and tolerance [79]. When this immune balance is disrupted, it can contribute to complications such as miscarriage, preeclampsia, or premature labour. Glucocorticoids can regulate the uterine immune system by directly or indirectly acting on these immune cells. Studies involving conditional ablation of GRs in mouse uteri have suggested the essential role of endometrial glucocorticoid signalling in recruiting appropriate immune cells [159]. In addition, in these GR knockouts, the expression of genes required for implantation was altered. This implies that the glucocorticoid signalling in the uterus might play a critical role in establishing the molecular groundwork that determines uterine receptivity [159].

Administration of exogenous glucocorticoids affects the number and function of uterine immune cells [160]. In the initial stages of pregnancy, uterine natural killer (uNK) cells control trophoblast invasion and changes in the vascular functions that help placental growth [161,162]. GRs are expressed in uNK cells, and their cytotoxic activity is responsive to cortisol treatment [163]. The administration of prednisolone in women experiencing recurrent miscarriage correlates with decreased uNK cell counts; however, there is yet to be a demonstrated connection between this glucocorticoid-induced reduction in uNK cells and the success of the pregnancy [164]. Macrophages exist in two distinct forms: an M1, or pro-inflammatory phenotype, and an M2 phenotype associated with tissue remodelling, immunosuppression, and angiogenesis [165]. During the peri-implantation period, endometrial macrophages lean towards the M1 phenotype and shift towards the M2 phenotype post-implantation [166,167]. GRs are expressed in human endometrial macrophages during the secretory phase, which coincides with the uterine receptivity period, implantation, and the menstrual phase, as well as at full term, but the function of glucocorticoid signalling in these macrophages remains largely unexplored [168,169]. Dendritic cells and regulatory T cells play a vital role in determining whether the blastocyst is accepted or rejected. However, the physiological impacts of glucocorticoids on these cell types within the endometrium remain under-researched [170].

In the peripheral immune system, glucocorticoids exhibit a broad range of direct influences on the development and function of immune cells, which can potentially affect the composition of immune cells available for recruitment to the reproductive tract [19]. They restrict the formation of immature dendritic cells from monocytes and inhibit the maturation of immature dendritic cells in vitro [171]. Elevations in serum glucocorticoid levels have been linked to the rapid increase of mature NK cells in peripheral circulation in vivo [172]. Furthermore, glucocorticoids stimulate substantial apoptosis in T and B cells, mature dendritic cells, basophils, and eosinophils [20]. Glucocorticoids suppress the transcription of pro-inflammatory cytokines and chemokines in macrophages and guide macrophages towards the M2 phenotype [160]. Dexamethasone can also trigger components of the inflammasome, thereby enhancing the innate immune response in macrophages [173]. The response of the immune system to glucocorticoids relies on both the physiological or stress-induced hormone levels and the duration of the stimulus. This suggests that the immune response that oversees reproduction is partly managed through physiological glucocorticoid signalling by the GR and is susceptible to hormone fluctuations.

#### GR Isoforms in the Uterus

GR is found in the uterus, predominantly within the stromal fibroblasts, lymphocytes, and endothelial cells, with only minimal presence in the glandular epithelium. Contrary to the cyclical changes reported with the estrogen receptor (ER) and progesterone receptor (PR), GR expression remains steady throughout the menstrual cycle [174]. During labour, there is a significant increase in total GR and GRα protein levels, while the levels of GRβ do not undergo any change [175]. Given the limited research on the expression of GR isoforms in the uterus, further studies are necessary to investigate if differential expression of these isoforms is behind the contradictory effects of glucocorticoids reported in the uterus.

## 8. Pregnancy and Parturition

During the second and third trimesters of pregnancy, the placenta begins to produce CRH, which has the same biological activity as the CRH produced by the hypothalamus (Figure 5) [99]. The placental CRH (pCRH) is secreted into both the maternal and foetal compartments. Unlike its inhibitory effect on the CRH-producing cells of the hypothalamus, cortisol stimulates the synthesis and release of pCRH [99], leading to a shift from negative feed-back regulation of the maternal HPA axis to a positive feed-forward cycle. Consequently, the circulating CRH levels in pregnant women are 1000 to 10,000 times higher than those of non-pregnant individuals [176]. Throughout gestation, most of the CRH is inactively bound to the CRH-binding protein (CRH-BP); however, during the third trimester, a spike in CRH occurs without a corresponding increase in CRH-BP levels in the maternal circulation, leading to an increase in the levels of active CRH [177]. It has been proposed that this rise in CRH levels may determine the length of gestation and initiate uterine contractility at the start of labour, acting as a “placental clock” [176]. However, if certain abnormalities occur during pregnancy, such as preeclampsia or maternal stress, these alterations may occur prematurely resulting from excess cortisol, which may lead to adverse maternal and/or foetal outcomes, including preterm birth and foetal growth restriction [99].

During pregnancy and parturition, uterine contractility can be separated into at least four stages. The first stage, known as phase 0, is the pregnancy period, where the uterus stays relatively quiescent through the action of inhibitors of uterine contractility such as progesterone [179]. In the first phase of parturition (phase 1), the uterus responds to mechanical stretch or uterotrophic priming by activating a set of genes essential for myometrial contraction [179]. As parturition enters phase 2, the uterus is stimulated by various uterotonins such as prostaglandins (PGs), oxytocin, and CRH [179]. The third and final phase involves uterine involution following the delivery of the foetus and placenta, a process primarily influenced by oxytocin [179]. Glucocorticoids help with two major events that lead to parturition [180]. Firstly, glucocorticoids stimulate the production of the placental enzyme cytochrome P450 17α hydroxylase (P450c17), which converts C21 steroids (pregnenolone and progesterone) into C19 steroid aromatase precursors [181]. This triggers a decrease in progesterone along with an increase in estrogen.

The rising estrogen then primes the myometrium from a state of quiescence to a contractile state by upregulating the expression of contraction-associated proteins (CAPs) such as connexin 43 [Cx43], oxytocin receptor (OTR), and prostaglandin (PG) receptor. After myometrium priming, sufficient uterotonins, such as prostaglandins, are synthesised to complete the feed-forward process of parturition [181]. Apart from initiating myometrial contractions, prostaglandins E2 and F2α (PGE2 and PGF2α) also facilitate cervical ripening, membrane rupture prior to parturition, and promote the expression of placental P450c17 [181]. In humans, prostaglandin production is compartmentalised within the tissues of the pregnant uterus [182]. While chorion and decidua also contribute, amnion is predominantly responsible for the formation of PGE2, and its output surges at labour onset [182]. Glucocorticoids have been found to elevate PGE2 and PGF2α synthesis in the amnion and chorion by inducing the expression of prostaglandin H synthase (PGHS-2), an enzyme involved in the rate-limiting step in prostaglandin synthesis [183,184]. Furthermore, PGE2 and PGF2α enhance 11β-HSD-1 and decrease 11β-HSD-2 activity in chorion, leading to increased cortisol production [185]. These feedback loops serve to increase both local cortisol and local PG concentrations [185]. 15-Hydroxyprostaglandin dehydrogenase (PGDH), an enzyme present in the chorion, metabolises PGs generated from the amnion and the chorion and prevents their passage to underlying tissues [182]. Its activity is usually decreased in preterm births, and as a result of the reduced metabolic barrier, the PGs are thought to prematurely stimulate myometrial contractility. Glucocorticoids have been shown to decrease the expression and activity of human chorionic PGDH [182].

Apart from stimulating the synthesis and inhibiting the metabolism of PG, glucocorticoids can also promote the rupture of the amnion. Apoptosis of the amnion epithelial layer occurs before the rupture of the foetal membrane, a key factor for both term and preterm delivery. Studies on human amnion epithelial cells in ex vivo cultures have demonstrated that cortisol triggers this apoptosis through the extrinsic apoptotic pathway [186]. Cortisol also reduces the quantity of collagen proteins and lysyl oxidase, an enzyme responsible for collagen cross-linking, in amnion cells through a process called lysosome-mediated autophagy [187]. The tensile strength of the amnion is determined by collagen content, and a reduction in collagen content has been observed in women experiencing preterm membrane rupture [188]. Therefore, glucocorticoids’ degenerative impact on amnion cells could be a contributing factor to the premature rupture of foetal membranes, leading to preterm labour.

Placental CRH also directly influences the uterus and cervix by increasing the changes instigated by estrogen within these structures. It stimulates dehydroepiandrosterone (DHEA) production from the foetal adrenal gland, which is required for the rise in placental estrogen needed to allow for the increase in uterine oxytocin and prostaglandin receptors, which allow the onset of parturition [189,190]. At the same time, cortisol upregulates prostaglandin production from the placenta and foetal membranes to induce membrane rupture and enhance contraction [190].

Numerous studies have been conducted looking at the impact of externally administered glucocorticoids on inducing parturition. In humans, intra-amniotic glucocorticoids can induce labour, whether at >41 weeks gestation with 20 mg of betamethasone [191] or in patients who are >12 days past their expected date of delivery and were given 500 mg of hydrocortisone [192]. In pregnancies involving triplets or quadruplets, maternal administration of betamethasone led to an increase in uterine contractions, premature labour, and alterations in the cervix [193]. Furthermore, the average gestation period in a cohort of term-born infants was reduced by an average of 4 days in pregnant women who received glucocorticoid treatment (a single dose of 12 mg of dexamethasone or betamethasone given intramuscularly twice over a 24 h period) during their 30th week of pregnancy as opposed to those who did not undergo glucocorticoid treatment [194]. Another concern relates to the excessive administration of synthetic glucocorticoids to stimulate foetal lung maturation in women at risk of preterm labour [195]. Despite numerous beneficial effects of endogenous glucocorticoids, such as foetal organ maturation [195], exogenously administered corticosteroids given to pregnant women at risk of preterm labour [193] and to animals [196] have been demonstrated to stimulate uterine activity. Thus, it is clear that both term and preterm births are a result of processes leading to an increase in PG production, with glucocorticoids playing an important part.

### Effect of Excess Cortisol on Myometrial Contractility during Parturition

Increased levels of cortisol due to maternal stress or an overactivated foetal HPA axis could lead to increased myometrial contractility and an increased chance of preterm birth.

Contrary to expectations, stress endured either prior to pregnancy or during gestation does not predict a preterm birth outcome, despite numerous studies suggesting an association. Women experiencing preterm labour often show significantly elevated CRH levels in comparison to controls at the same gestational stage [189,197]. Interestingly, these heightened CRH levels can emerge weeks ahead of preterm labour onset [179]. Under normal circumstances, cortisol levels increase towards the final stages of gestation and during the initiation of labour [135,189,197]. However, maternal stress and high cortisol levels can lead to the activation of labour-related mechanisms in the placental and foetal membranes, potentially triggering labour prematurely [189,197]. Understanding how stress can induce preterm labour in some women but not others may be related to the expression of different GR isoforms. The differential expression of GR isoforms between individuals may contribute to the risk of preterm delivery under stressful conditions, but this needs to be investigated further.

## 9. Foetus

During pregnancy, the foetus relies heavily on glucocorticoids for appropriate development. The concentrations of cortisol required by the foetus change as pregnancy advances, with the placenta allowing increasing levels of glucocorticoid transfer as the foetus approaches term. Indeed, maternal glucocorticoids contribute to the growth, development, and survival of the foetus in all mammals [198]. Of particular clinical importance is the role of glucocorticoids in ensuring appropriate lung maturation, but they are also vitally important for the development of the kidney, heart, adrenal, genital, and almost all other foetal tissues [199]. However, exposure to high levels of cortisol due to maternal stress can have significant consequences for foetal development [198]. While some of the negative effects of elevated maternal glucocorticoids are caused by disruption of placental formation [200,201,202], they can also directly impact the formation of key systems, including the cardiovascular system, kidneys, brain, and immune system [203,204,205]. Consequently, it can elevate the risk of future health issues such as cardiovascular, kidney, neurological, respiratory, and metabolic disorders, along with the likelihood of allergies, in the child’s later life [206]. Synthetic glucocorticoids are also commonly administered to women who are at risk of giving birth prematurely, as well as those with certain medical conditions like asthma, systemic lupus erythematosus, and hyperemesis gravidarum [207]. Excessive exposure to synthetic glucocorticoids during gestation can have far-reaching impacts on an individual’s health throughout their lifespan. Previous reviews have focused on describing the wide-ranging, programmed effects of glucocorticoids [207]. Below are a few examples. Elevations in maternal glucocorticoids have been shown to program an altered stress response [208], motor developmental delays, and higher blood pressure [209].

Studies on rodents have also shown that administering synthetic glucocorticoids during pregnancy can lead to sex-specific changes in placental development [201,210] and result in negative health outcomes for offspring [211,212]. For example, dexamethasone administration in mice led to reduced placental size and reduced foetal growth in female placentas [211]. Similarly, in sheep studies, exogenous administration of betamethasone led to reduced foetal growth [211], increased placental apoptosis [213], and decreased levels of insulin-like growth factors [213], suggesting that excess maternal glucocorticoids can impact foetal growth and nutrient transport through changes in the placenta.

### 9.1. Sex Differences

There is a large body of evidence indicating that the effects of prenatal stress on offspring differ between males and females [214]. For example, in asthmatic mothers, increased cortisol levels and decreased cortisol metabolism by 11β-HSD2 in the placenta were associated with reduced birth weights and suppressed foetal adrenal function only in female foetuses [215]. Animal studies have also shown that maternal stress or exposure to cortisol can cause sex-specific changes in placental dysfunction [200,216] and negative health outcomes for offspring [216,217]. The female placenta appears to adapt to fluctuations in glucocorticoid levels, as indicated by alterations in cortisol metabolism, placental cytokine expression, IGF axis signalling, adrenal functionality, and growth [218,219]. In contrast, the male placenta seems less responsive to changes in glucocorticoid levels, with no changes in cortisol-regulated pathways [217]. This lack of adaptation in the male placenta is associated with a higher likelihood of intrauterine growth restriction, preterm birth, or stillbirth [219]. On the other hand, female placentas adapt in a way that may lead to reduced growth but enhanced survival [217].

The female placenta has been observed to potentially allow greater glucocorticoid exposure due to the altered activity of enzymes responsible for cortisol metabolism, including 11β-HSD1 2 [218,220]. Intriguingly, unique transcriptional profiles emerge from the differential expression of GR isoforms in response to maternal dexamethasone exposure, and this expression varies between male and female mouse placentas [44,221].

Research also suggests that the impacts of prenatal maternal stress are not only sex-specific but can also extend beyond the neonatal period [222]. For instance, exposure to stress during foetal life in females has lingering effects into early childhood and adolescence, as evidenced by increased anxiety levels, compromised executive function, and neurologic markers linked to these behaviours [223]. Early exposure to high maternal cortisol levels in pregnancy led to a significantly larger amygdala and increased anxiety levels in female offspring, but not in males [223]. Additionally, maternal stress correlated with negative emotionality in female offspring, but this association was not found in males [224]. This evidence suggests that exposure to stress in early life can lead to sex-specific outcomes in foetal programming. Such programmed disease outcomes have even been shown to have impacts that persist in the next generation [225,226]. Thus, to understand how excess cortisol or synthetic glucocorticoids affect foetal development and mediate sex-specific changes in foetal outcome and programming, it is important to study the glucocorticoid receptor and its downstream signalling pathways [214,227].

### 9.2. Glucocorticoid Receptor Isoform in the Placenta

There are thirteen glucocorticoid receptor (GR) protein isoforms in the placenta of humans [222,223], guinea pigs [224], sheep [225], mice [141], and rats [226] that vary in relation to gestational age at delivery, foetal sex, circulating glucocorticoid concentration, foetal growth, and maternal stress. Changes in GR isoforms in the placenta may be a novel mechanism to explain how different placentae have varied responses to pregnancy complications, including pregnancies complicated by asthma. For instance, an increase in cortisol levels is associated with a rise in the expression of GRβ protein in the cytoplasm of male placentae in pregnancies complicated by asthma and in both the cytoplasm and nucleus of placentae in small for gestational age (SGA) pregnancies [222]. Thus, in scenarios of increased cortisol exposure, males may establish a state of glucocorticoid resistance through the increased expression and nuclear localisation of GRβ, which then suppresses the activity of GRα in a dominant-negative manner [222]. This resistance state might also be influenced by the nuclear co-localisation of GR-A and GR-P in the male placental trophoblast [222].

The interaction between different GR isoforms could also lead to the activation of different downstream pathways and target genes. For example, under basal conditions, GRα and GRβ are co-expressed in neutrophils [227]. However, when these cells are exposed to IL-8, there is an increase in GRβ, which subsequently leads to a decrease in corticosteroid-induced apoptosis [227]. Similarly, an increase in the GRβ: GRα ratio in male foetuses from pregnancies complicated by asthma could potentially allow these male foetuses to continue growing despite a high glucocorticoid environment, by upregulating genes related to growth and preventing apoptosis [222]. Unlike male foetuses, female foetuses have reduced growth in high-glucocorticoid environments [209]. In SGA female placentae, there is a decrease in the nuclear expression of GRβ [222]. This change is coupled with an increase in the nuclear expression of GRα-D3 and an increase in the cytoplasmic expression of GRα-C [222]. The interaction of GRα-A with GRα-C or GRα-D3 could contribute to maintaining glucocorticoid sensitivity in female placentae [222], potentially inducing higher rates of apoptosis, thereby contributing to reduced foetal growth. In preterm placentae, males displayed elevated cytoplasmic levels of GRα-C and GR-A, while female placentae showed increased nuclear levels of GRα-C compared to term placentae [223]. GRα-C is recognised as a powerful activator of glucocorticoid-triggered apoptosis, a process that includes the stimulation of the mitochondrial BIM pathway, consequently leading to the activation of caspases 9 and 3, resulting in cell death [228]. The placenta of preterm pregnancies has been shown to exhibit increased expression of BIM and BAD [229]. This upregulation of GRα-C might be crucial in understanding the pathophysiology of preterm birth. It is possible that an escalation in glucocorticoid-induced cell death, mediated by GRα-C, could be a precursor to preterm labour [209]. These findings suggest that in a high glucocorticoid environment, responsiveness to glucocorticoids hinges on the presence or absence of different GR variants, as well as the concurrent expression and interaction between different GR isoforms within placental tissue.

## 10. Conclusions

Glucocorticoids play a central role in regulating female reproduction and can act as a mediator of immune suppression or an activator of inflammation, depending on the GR isoform profile. GRs have been identified in many tissues and cell types of the reproductive tract and have a central role in pregnancy. However, the lack of data identifying the expression of different GR isoforms and their role in regulating the reproductive cycle is an area that requires further investigation.

## Figures and Tables

**Figure 1 biology-12-01104-f001:**
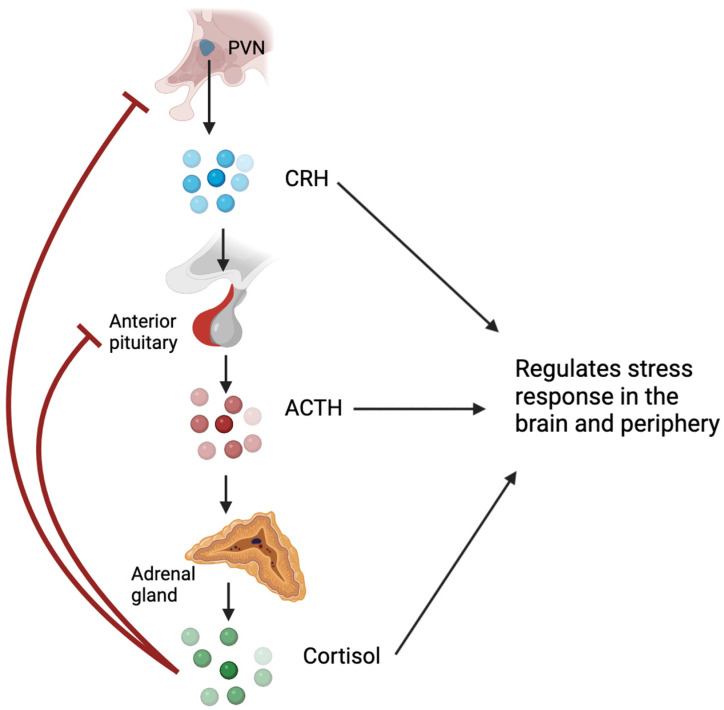
Hypothalamic–pituitary–adrenal axis. Cells in the paraventricular nucleus produce CRH in response to physical or psychological stress. CRH binds to cells in the anterior pituitary gland to stimulate the production of ACTH, which acts on the adrenal glands to increase the production of glucocorticoids, which, in turn, regulates the stress response. Glucocorticoids can also exert their negative effects on CRH and ACTH via a negative feedback mechanism. PVN—paraventricular nucleus; CRH—corticotrophin-releasing hormone; ACTH—adrenocorticotrophic hormone. Created on bio render. Adapted from [6].

**Figure 3 biology-12-01104-f003:**
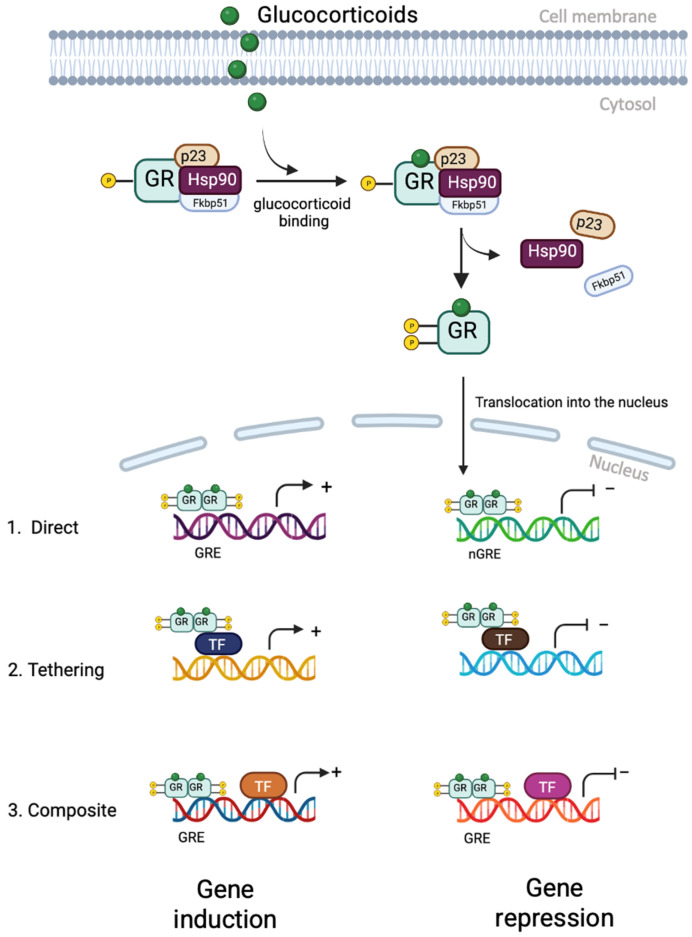
Genomic action of a glucocorticoid receptor (GR). The GR binds to glucocorticoids in the cytoplasm, leading to the dislocation of its chaperone protein complex and translocation into the nucleus. In the nucleus, the GR can either directly induce or repress gene expression by binding to GREs or nGREs, respectively. It can also interact with other TFs by tethering or composite actions. GR—glucocorticoid receptor; Hsp90—heat shock protein 90; Fkbp51—FK506-binding protein 51; GRE—glucocorticoid response element; nGRE—negative GRE; TF—transcription factor. Created on bio render. Adapted from [45].

**Figure 4 biology-12-01104-f004:**
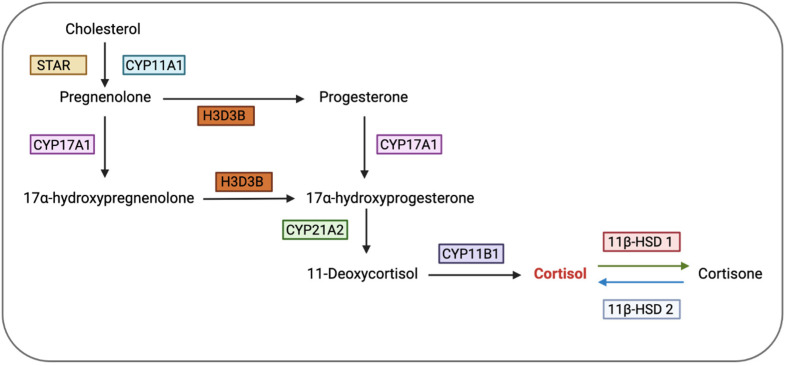
Steroidogenesis pathway, illustrating the key factors involved in the conversion of cholesterol into progesterone and cortisol. STAR, steroidogenic acute regulatory protein; CYP11A1, cytochrome P450 family 11 subfamily A member 1; CYP17A1, cytochrome P450 family 17 subfamily A member 1; HSD3B, hydroxy-delta-5-steroid dehydrogenase, 3 beta- and steroid delta-isomerase; CYP21A2, cytochrome P450 family 21 subfamily A member 2; CYP11B1, cytochrome P450 family 11 subfamily B member 1; HSD11B1, hydroxysteroid 11-beta dehydrogenase 1; HSD11B2, hydroxysteroid 11-beta dehydrogenase 2. Created on bio render. Adapted from [111].

**Figure 5 biology-12-01104-f005:**
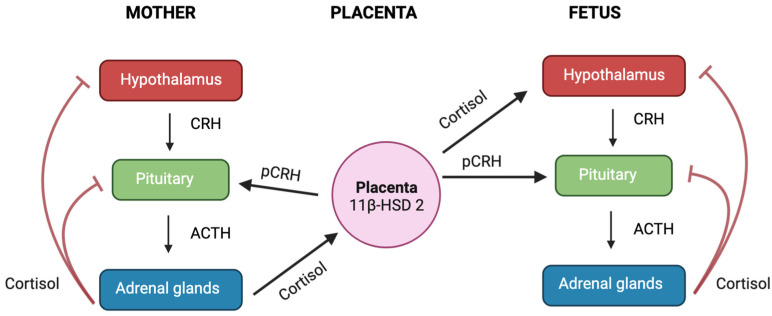
Stress axis during pregnancy. Under stress, the maternal HPA axis produces cortisol, which can act on the placenta to stimulate the production of pCRH. Most of the cortisol is metabolised by placental 11β-HSD 2, with 20% crossing over to the foetus. Increased levels of cortisol crossing over to the foetal compartment can over-activate the foetal HPA, leading to long-lasting effects on its growth and development. CRH—corticotrophin releasing hormone; ACTH—adrenocorticotropic hormone; 11β-HSD 2—type 2 11-beta hydroxysteroid dehydrogenase; pCRH—placental CRH; HPA—hypothalamus—pituitary—adrenal. Created on bio render. Adapted from [178].

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
