# Peer review of "Glucocorticoids and Their Receptor Isoforms: Roles in Female Reproduction, Pregnancy, and Foetal Development"

_biology, 2023, doi:10.3390/biology12081104_

Round 1

Reviewer 1 Report

In the submitted MS, the authors present a narrative review of the function of glucocorticoids in the reproductive function, particularly in the female reproductive organs and in pregnancy. The authors present a comprehensive review of the topic, often descriptive. Still, they discuss the available information and identify gaps in knowledge that deserve to be explored. In general, the paper is clear and illustrated with some figures summarizing some main mechanisms or information. Some of them may be discarded, as they are too superficial.

I am not sure the title of the paper in fact reflects its content. The duality of action of glucocorticoid in female reproduction and pregnancy, albeit hinted at some points of the MS, does not translate for the entire review. Authors should ponder to change the title of their paper to better match its content

Even though the authors state "This paper explores how stress and the body's reaction to it can affect fertility, reproduction, and fetal development" in the simple summary section, the paper focuses mainly on the glucocorticoid effects and the physiological effects of cortisol/glucocorticoid axis on the female reproductive function. Not on the effects of stress or stressors stimuli. This issue needs the authors' attention.

The abstract needs to be remodeled to briefly identify the type of review and to present the methods used for the retrieval of papers used in the review. it could also benefit from the precocious identification of the type of paper it represents (a review, not a research report).

Section 2 (Immunological and therapeutical function of glucocorticoids) does not match the scope of the review, nor the content of the MS as it is. Please remove it. If the authors consider that part of the content of this section is important to understand other sections, then move the needed information to that spot.

Change the order of appearance of sections 3 and 4. Presenting the receptors closer to receptor signaling seems to follow a more logical order. the section on Bioavailability can appear before the  "receptors" section, or even be integrated within one of the sections regarding signaling or receptors topics

Compared with the genomic effects of glucocorticoids, the non-genomic effects are poorly explored. Can the authors provide more detail on the non-genomic pathway of glucocorticoid action? In some species, they seem to be involved with progesterone signaling in the uterus or placenta, so it would be of relevance for this review.

Regarding the "Glucocorticoid receptor isoforms in the ovary" sub-section, some repetition of information exists. Please revise the text and reduced any duplications.

Some minor issues include:

- Figures 1 and 5, and the respective captions, need to be revised.

In the Figure 1 caption, a mention of a minus signal (-) exists in the caption that is not represented in the image. It might be transposed into the red truncated lines (they are often used to illustrate inhibitory actions).

In Figure 5, the caption states "Increased levels of cortisol in the fetal compartment can over activate the fetal HPA leading to long-lasting effects on its growth and development". However, those effects are not represented in the image, as far as I understand the lines: the red truncated lines would signify a negative action or the inexistence of an action, do they not?

- Please check the numbering of the sections and subsections. Some incongruencies were found

- Revise the reference list:

a. capitalize the first letter in the journal name

b. do not capitalize the entire name of authors; only the first letter of the name

Reviewer 2 Report

This manuscript entitled “Deciphering the Dichotomy: The Role of Glucocorticoid Receptor Isoforms in Cortisol´s Paradoxical Impact on Reproduction and Pregnancy”, systematically examines the involvement of glucocorticoid receptors throughout reproduction, pregnancy and parturition, including the placenta. It gives us a very comprehensive overview of the effects of stress on female fertility and the course of pregnancy in these circumstances. 

However, I have some minor concerns about the presentation of the review: 

1) Figure 1 shows an inhibitory effect on CRH and ACTH molecules, which I believe is only on the producing tissues, in this case the PVN and the anterior pituitary, as shown in Figure 5.

2) In the footer of figure 1 it is mentioned that it is based on reference number 36, which has nothing to do with what the figure represents, it seems rather to be an adaptation of reference 6. 

3) Regarding figure 2, this is very similar to the original in format, color and idea, so publishing this figure requires applying for copyright. I therefore propose that a change be made in the format of the same as well as the colors where no permission is required from the authors of reference 33.

4) Have been not reported any new isoforms since 2016? Could you updated this information in the figure.

5) Check in the  figure 3, it says that the adaptation corresponds to the reference 52 which is not related since this reference is the characterization of the promoter of the GNRH .  The related reference that in my opinion is the reference 36, that explains these mechanisms. 

 6) Write the reference 50 in lowercase because all authors appear in uppercase.

Round 2

Reviewer 1 Report

In the revised MS the authors responded to my concerns. Some minor suggestions remain in the main text, mainly for clarity of the speech (for detail, address the commented copy of the MS attached to this report).
I would further suggest some changes regarding subsections 7.1.2 and 7.2.1  regarding the location of GR isoforms in the ovary and uterus, respectively: move it to before presenting the glucocorticoids actions/functions in those organs. It would allow for a more coherent flow of information.

Also, on line 379 - the reference Jeon et al is wrongly numbered (possibly due to the text changes occurring during the revision process. To be sure that it did not happen with another reference, check across the MS

Line 502 - "The expression levels of GR in the follicle and corpus luteum remains unchanged during the stages of follicular maturation and ovulation in rats" - the events here respect only the follicular development. remove the words "corpus luteum" from the sentence

regarding your question about the capitalization of the journal names, revise the reference list and apply the same style used for reference 1
